# Competition between Hydration Shell and Ordered Water Chain Induces Thickness-Dependent Desalination Performance in Carbon Nanotube Membrane

**DOI:** 10.3390/membranes13050525

**Published:** 2023-05-18

**Authors:** Siyi Liu, Liya Wang, Jun Xia, Ruijie Wang, Chun Tang, Chengyuan Wang

**Affiliations:** 1Faculty of Civil Engineering and Mechanics, Jiangsu University, Zhenjiang 212013, China; 2212023014@stmail.ujs.edu.cn (S.L.); xiajun@ujs.edu.cn (J.X.); wangruijie@ujs.edu.cn (R.W.); tangchun@ujs.edu.cn (C.T.); 2Zienkiewicz Centre for Computational Engineering, Faculty of Science and Engineering, Swansea University, Bay Campus, Swansea SA1 8EN, UK

**Keywords:** thickness effect, carbon nanotube, nanoporous monolayer graphene, ion dehydration, desalination performance

## Abstract

Exploring new reverse osmosis (RO) membranes that break the permeability-selectivity trade-off rule is the ultimate goal in seawater desalination. Both nanoporous monolayer graphene (NPG) and carbon nanotube (CNT) channels have been proposed to be promising candidates for this purpose. From the perspective of membrane thickness, both NPG and CNT can be classified into the same category, as NPG is equivalent to the thinnest CNT. While NPG has the advantage of a high water flux rate and CNT is excellent at salt rejection performance, a transition is expected in practical devices when the channel thickness increases from NPG to infinite-sized CNTs. By employing molecular dynamics (MD) simulations, we find that as the thickness of CNT increases, the water flux diminishes but the ion rejection rate increases. These transitions lead to optimal desalination performance around the cross-over size. Further molecular analysis reveals that this thickness effect originates from the formation of two hydration shells and their competition with the ordered water chain structure. With the increase in CNT thickness, the competition-dominated ion path through CNT is further narrowed. Once above this cross-over size, the highly confined ion path remains unchanged. Thus, the number of reduced water molecules also tends to stabilize, which explains the saturation of the salt rejection rate with the increasing CNT thickness. Our results offer insights into the molecular mechanisms of the thickness-dependent desalination performance in a one-dimensional nanochannel, which can provide useful guidance for the future design and optimization of new desalination membranes.

## 1. Introduction

Seawater desalination is one of the most promising solutions to fresh water shortages all over the world [1,2,3,4]. As the most common method of desalination, state-of-the-art reverse osmosis (RO) technology uses a semipermeable membrane to filter out salt ions but allow water passage [5,6]. Novel filtration membranes with substantially improved permeability, selectivity, chemical stability, and resistance to fouling remain an open topic of intense interest as of now [7,8,9].

With the rapid development of nanotechnology, the boom of novel two-dimensional (2D) materials demonstrates great promise for the renewal of traditional membranes [1,2]. Based on the dimension of the confined nanochannel, RO membranes can be roughly divided into three types: the membranes with quasi-one dimensional (1D) nanopores, e.g., nanoporous graphene (NPG) [10,11,12], the ones with 1D nanochannels such as carbon nanotube (CNT) [13,14,15,16], and those with 2D interlayer channels, e.g., stacked graphene oxide (GO) [17,18,19,20]. Due to their similar circular pore shapes, the membranes made of quasi-1D NPG and 1D CNT were compared in terms of their liquid transport performance [11]. To maintain a balance between permeability and selectivity, novel designs based on large diameters, including chemical modifications on the pore edge, or a rotating NPG cylinder have been extensively adopted and yielded satisfactory results [15,21].

Although the quais-1D NPG and 1D CNT have different dimensions, they can be considered similar in terms of membrane thickness since NPG can function as the thinnest CNT. The thickness effect can therefore be employed to measure the desalination differences between NPG and CNT. Considering the advantages of quasi-1D NPG’s high water flux rate and 1D CNT’s excellent salt rejection capacity, it is reasonable to hypothesize that a critical thickness value might exist, which enables the combination of the advantages of both NPG and CNT and thus makes the CNTs the optimal membrane candidates. Previous molecular dynamics (MD) simulations demonstrated that a 2.34 nm-thick CNT membrane can exhibit nearly doubled water permeability while maintaining 100% NaCl rejection [22]. This is inspiring, as it suggests that desalination properties could be easily doubled by simply choosing a proper CNT thickness without further ado. Compared with the commonly used diameter variable in optimizing desalination performance, not enough attention has been paid to the thickness effect. Up to now, the only thing clear for us is that such a thickness effect arises from the dehydration of salt ions [22]. A further investigation is thus urgently required to uncover the underlying mechanisms of the observation at the atomic scale.

The hydration state of salt ions within the nanochannel is the key to analyzing many biological and chemical processes, such as the selective permeability of cellular membranes, the ion selectivity of filtration membranes, etc. [23,24,25]. Note that the specific hydration configuration of salt ions within CNT also plays a pivotal role in the analysis of the thickness effect. However, the transient penetrating process makes it extremely difficult to capture the necessary details and understand how the specific water configurations contribute to thickness-dependent ion dehydration. In this work, we mimic the ion penetrating process by artificially tethering a Na^+^ ion with a spring, which is allowed to move freely in the nanopores. Through comprehensively examining the structure properties of salt ions and water molecules within the two hydration shells, we find that the thickness effect of CNT is induced by the competition between the hydration shell and the ordered water chain structure. In particular, the CNT membrane exhibits optimal desalination performance when its thickness is sufficiently large to accommodate two hydration shells of salt ions. Meanwhile, water molecules within hydration shells have to be compatible with the ordered water chain structure. The resulting steady and further confined ion path contributes to the unchanged dehydration degree, which is consistent with the finally unchanged dehydration energy barrier reported in the literature [22].

## 2. Simulation Details

The schematic of the simulation system is illustrated in Figure 1. The feed side with NaCl solution and the permeate side with pure water are separated by a CNT, which is enclosed by two graphene sheets with appropriate holes. A net concentration of 0.5 mol/L is yielded with 31 Na^+^ and 31 Cl^−^ randomly placed in 3432 water molecules. 607 water molecules are accommodated on the permeate side. To simulate the liquid transport through CNT, different pressures are exerted on the graphene pistons on the feed and permeate sides, and a pressure difference ΔP = P_feed_ − P_permeate_ is thus rendered.

As shown in Figure 1b, different thickness values from L_1_ to L_22_ are adopted, where the subscript *i* in *L_i_* represents the number of carbon rings. Values of *L*_1_, *L*_4_, *L*_7_, *L*_10_, and *L*_22_ CNTs are 0.34, 0.709, 1.078, 1.447, and 2.923 nm, respectively.

Water molecules are described using the SPC/E model [26], and the interactions for all other atomic species are modeled using Lennard-Jones (LJ) potential with a Coulombic term. All the carbon atoms are held rigid. Parameters for carbon and salt ions are adopted from a previous paper, and Lorentz-Berthelot mixing rules are used for cross-interactions [27]. The long-range Coulomb interaction is calculated using a particle-particle particle-mesh (PPPM) [28] algorithm with an accuracy of 10^−4^.

MD simulations are carried out using the LAMMPS package to predict thickness-dependent desalination performances [29]. Simulations are carried out in the constant-volume and constant-temperature (NVT) ensemble at 300 K using a Nose-Hoover thermostat. A time step of 1.0 fs is used for the velocity-Verlet integrator. Periodic boundary conditions are applied in the x and y directions, and the corresponding box sizes are Lx = 31.98 Å and Ly = 34.08 Å. Box length in the z direction depends on the length of the nanochannel.

The first step is to set the pressures on both pistons on the feed and permeate sides to the ambient pressure of 0.1 MPa. The desalination system is then allowed to relax for 5 ns under these conditions. After the system reaches equilibrium, the desired pressure of 400 MPa is applied to the piston on the feed side, and the P_permeate_ remains unchanged. As a commonly used strategy to control the simulation time in MD simulations, a higher pressure value than the practical desalination pressure is adopted to increase the computational efficiency [30,31]. The linearly increasing water flux with simulation time indicates that the simulation system reaches the steady state, as shown in Appendix A. All the simulations are run until 80% of the water molecules on the feed side are transported to the permeate side, and the corresponding water flux rate and NaCl rejection rate are calculated [32]. As commonly used in many works, NaCl rejection is defined as the 100 percent subtracting ratio of the ion concentrations in the permeate and feed sizes [22,30]. To maintain electrical neutrality, the number of permeating Na^+^ and Cl^−^ is almost the same. Hence, the NaCl rejection calculation is based on the permeating Na^+^ number.

## 3. Results and Discussion

Herein, desalination performance in (6, 6), (7, 7), (8, 8), and (9, 9) CNTs is studied, where the effective pore diameters after subtracting the van der Waals diameter of 0.34 nm are 0.472, 0.608, 0.743, and 0.879 nm, respectively. Water flux rate and NaCl rejection against diameter are shown in Figure 2a,b for *L*_1_ CNTs (NPG) and *L*_22_ CNTs. It is clearly seen that strong competition exists between salt rejection and water flux. Not surprisingly, NPG shows superiority in water flux rate, while CNTs exhibit advantages in salt rejection. Especially for the (7, 7) CNT, a significant increase from 51.5% to 100% is observed for the NaCl rejection when the nanochannel changes from the *L*_1_ NPG to the *L*_22_ CNT, while the corresponding water flux rate is found to decrease from 141.4 ns^−1^ to 94.8 ns^−1^. This makes (7, 7) CNT an ideal candidate to investigate the thickness transition from *L*_1_ to *L*_22_ CNT. Thus, the thickness-dependent desalination performance of (7, 7) CNTs is illustrated in Figure 2c. It is seen from the figure that with rising thickness, the two desalination parameters finally reach a stable stage after an initial transition zone where they change rapidly with the variation of CNT thickness.

To uncover the underlying mechanisms of thickness-dependent desalination performance, it is necessary to examine the detailed ion penetrating process. However, the instantaneous penetration behavior made it almost impossible to capture the necessary details. Within the limited range of CNT length considered, the ion penetrating process includes two successive and symmetrical steps: entering CNT from the bulk solution on the feed side and returning to the bulk solution on the permeate side. In view of the same hydration structure in bulk solution, unique ion hydration configurations within the nanochannels play a pivotal role in generating the thickness effect. Therefore, in this work, one single Na^+^ ion was artificially placed in the pore center and tethered with a spring to allow free motion within nanopores. In real-life cases, the ions sometimes cannot pass through the CNTs when the thickness is too large or the diameter is relatively small. We, however, keep utilizing this protocol throughout our analyses as it enables us to explain the desalination difference from a distinctive perspective.

The number density map of ion distribution within CNTs is provided in the top row of Figure 3. When a Na^+^ ion passes through the *L*_1_ NPG membrane, its relative affinity with the pore edge is manifested by the loose distribution pattern. With the increase in membrane thickness, the Na^+^ ion was gradually restricted to the central region. When the membrane thickness is increased to *L*_7_, the Na^+^ ion has the largest probability of occupying the pore center. This trend is maintained when the membrane thickness is further increased to *L*_10_. Considering the ion-oxygen interaction, it is reasonable to speculate that the position shift of the Na^+^ ion is related to the water molecules within hydration shells. Hence, the hydration shell configurations of Na^+^ in different CNTs are studied, as shown in the bottom row of Figure 3. For the *L*_1_ NPG, water molecules within both hydration shells are completely disordered. Compared with the spherical hydration shells in bulk solution, water molecules within the graphene plane are expelled. The Na^+^ ion tends to be close to the edge of graphene, accompanied by one coplanar water molecule within the first hydration shell. When the membrane thickness is raised to L_4_, water molecules within the first hydration shell start to exhibit a two-column configuration, and a similar configuration always remains in thicker CNTs. Due to the pore diameter confinement, water molecules in the second hydration shell refer to the ones connecting to the two water columns in the CNT axial direction. Water molecules in the second hydration shell are also disordered in L_4_ CNT. With the further increase in membrane thickness, L_7_ CNT is long enough for water molecules in both hydration shells to form an ordered two-column chain structure. Held tightly by the ordered water chain structure, the motion area of Na^+^ ions is strictly restricted to the central region, as shown by the highest distribution density in the pore center of L_7_ and L_10_ CNTs.

Next, we examine the orientation of water molecules within hydration shells. Here, *α* is used to represent the angle between the oxygen-ion vector and the water dipole vector. Its probability distribution within the first and second hydration shells is plotted in Figure 4. Due to the strong electrostatic interaction between the oxygen atoms and the positive charge that the Na^+^ ion carries, the probabilities of *α* in the first and second hydration shells reach peak values at around 155° and 145°, respectively. With the increase in membrane thickness, *α* distribution probability becomes narrower and sharper. For the *α* probability in the first hydration shell in Figure 4a, the two-column patterned water molecules in the first hydration shell start to form in *L*_4_ CNT and exhibit highly ordered orientation in *L*_7_ CNT. A slightly higher peak value is observed in *L*_22_ CNT. As for the water molecules in the second hydration shell in Figure 4b, the discrepancy between the NPG and CNT membranes is clearly observed. The ordered second hydration shell starts to form in *L*_7_ CNT and is further strengthened with the increase in membrane thickness. The overall orientation distribution trend is consistent with the above analysis of hydration structures. To sum up, the ordered water chain structure within CNT can regulate the water configuration within the hydration shell. The competition between the ordered water chain structure and the hydration shell is the origin of the preference for ion position. It is easy to imagine that the narrow ion path in thicker CNTs would exert a larger resistance for ion passage in practical desalination applications, and thus a larger salt rejection would be rendered. Once the confined and concentrative path is formed in *L*_7_ CNT, the resistance remains unchanged with further increases in CNT thickness.

The ion hydration configuration within the nanochannels is closely related to their dehydration process. Using the aforementioned spring-tethered Na^+^ ion model in thickness-variant CNTs, the ion-oxygen radial distribution functions (RDFs) of Na^+^ in different CNTs are calculated, as shown in Figure 5a. The corresponding coordination numbers in the first (*Nc*_1_) and second (*Nc*_2_) hydration shells are summarized in Table 1. In accordance with the previous results, 5.6 and 17.1 water molecules were found in the first and second hydration shells [33,34,35]. The reduced number of water molecules in different CNTs is depicted in Figure 5b. When the Na^+^ ion enters the *L*_1_ NPG, about 4 water molecules are peeled from the second hydration shell, while the first hydration shell remains almost intact in spite of the deformation. Starting from the thinnest *L*_4_ CNT, dehydration occurs in both shells, and the reduced water number in the first hydration shell stabilizes at around 1.0. With the increase in CNT thickness to *L*_7_, the reduced water number in the second hydration shell rises from 10.0 to 12.7 and remains almost constant with the further increase in membrane thickness. This further confirms that the actual reduced number in ion dehydration is constant when the CNT membrane is thick enough to form a complete two-column water chain within the hydration shell. Despite the Na^+^ ion being artificially tethered within the nanochannel, the number of dehydrated water molecules can clearly manifest the dehydration energy barrier that the ion needs to overcome to squeeze through the narrow path inside the CNT. This conclusion is consistent with the unchanged dehydration degree above a certain thickness [22], as the input energy is not large enough to make the Na^+^ ion peel almost 13 water molecules from the second hydration shell to cross CNTs thicker than *L*_7_.

Generally speaking, this thickness effect applies to all the CNT diameters considered in this work, as shown in Figure 6. Similar to the case in (7, 7) CNT, intact hydration shells start to form at the *L*_7_ length in (6, 6) CNT. Nevertheless, the extreme confinement in *L*_1_ NPG could block almost all the ion passage. The increase in CNT thickness makes no significant contribution to salt rejection improvement but undermines the water flux rate instead, as shown in Figure 2a,b. For the (8, 8) and (9, 9) CNTs, water molecules exhibit four and six columns of chain structure, respectively [36]. As shown in Figure 6, the CNT length needed for intact hydration shells is larger than *L*_7_, and the position shift of Na^+^ ions originating from the competition between the hydration shell and ordered water chain can also be observed. Combined with the thickness-dependent salt rejection lines in Appendix A, the critical thickness values in (6, 6), (7, 7), (8, 8), and (9, 9) CNTs are *L*_1_ (0.34 nm), *L*_7_ (1.078 nm), *L*_10_ (1.447 nm), and *L*_10_ (1.447 nm), respectively. On the one hand, the diameter-dependence of the critical thickness value stems from the weaker confinement in large CNTs. On the other hand, in spite of saturated salt rejection in (8, 8) and (9, 9) CNTs, the actual salt rejection capacities are still inefficient and unsatisfactory. In practical desalination applications, the operating pressure is usually not higher than 10 MPa, which may loosen the restriction on the CNT diameter [30]. It has been found that the ideal size of the CNTs for desalination applications can be as large as 1.1 nm in diameter [37]. Hence, the (8, 8) and (9, 9) CNT membranes can be used for ideal salt rejection, and a larger critical thickness value may be expected in practical devices. Compared with the previously reported critical thickness of 2.34 nm [22], although differences exist in specific details, the nature of enhancing water permeation at the cost of ion rejection through proper choice of CNT thickness is exactly the same.

## 4. Conclusions

In our efforts to develop a novel desalination membrane that is expected to enjoy NPG’s high water flux rate and CNT’s excellent salt rejection capacity, we conducted MD simulations to examine the thickness-dependent desalination performance of CNTs. To this end, (6, 6), (7, 7), (8, 8), and (9, 9) CNTs with various thicknesses are used to construct the filtration membranes. Our simulation results show that, for the diameter ranges considered in this work, the salt rejection rate initially increases with increasing CNT thickness but levels off once the thickness exceeds a critical value. We found that CNT membranes demonstrate the most efficient desalination performance when the length of the CNT reaches a critical value that enables salt ions to form ordered hydration shells. Analysis based on molecular details revealed that the position preference of salt ions within the CNT nanochannel is the result of competition between the hydration shell and the ordered water chain structure. As the thickness of the membrane increases, the ion path through CNT narrows down and eventually becomes a confined route after the thickness approaches the critical value. In this process, the number of reduced water molecules from the hydration shells follows a similar trend of rising and then saturating, which shows a clear indicator of the dehydration energy barrier that ions have to overcome to squeeze through the CNTs with variable thickness. By revealing the rationale of the thickness effect from a unique perspective, we expect that the new findings of the present study can significantly deepen the understanding of ion transport through 1D nanochannels and provide theoretical guidance for practical desalination applications in the near future.

## Figures and Tables

**Figure 1 membranes-13-00525-f001:**
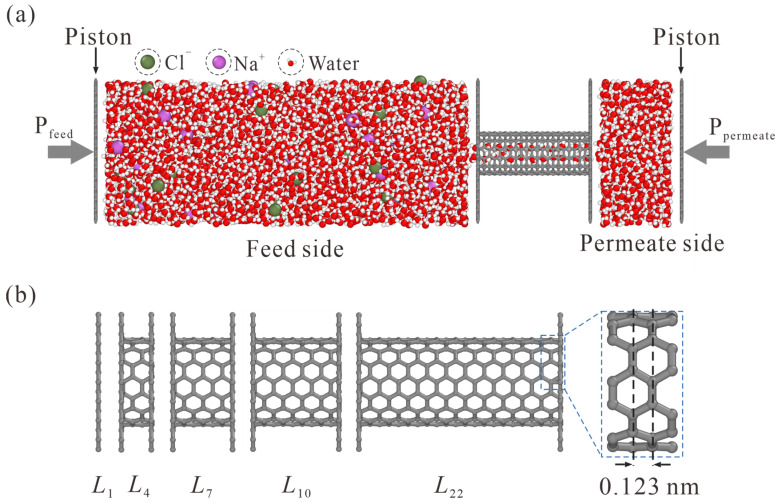
Schematic of the simulation model. (**a**) NaCl solution and pure water are placed on the feed and permeate sides, respectively. The pressure difference is exerted on graphene pistons to simulate liquid transport through CNT. (**b**) Side view of CNTs with different lengths. The subscript in *L_i_* represents the number of carbon rings. The *L*_1_ case is, in fact, a NPG monolayer.

**Figure 2 membranes-13-00525-f002:**
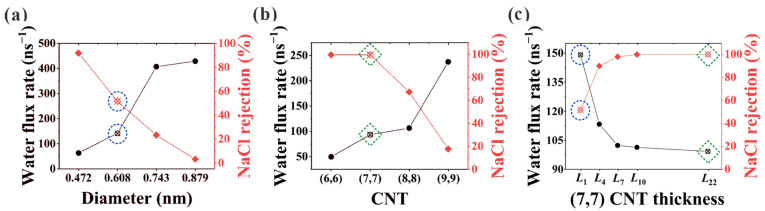
Water flux rate and salt rejection in (**a**) *L*_1_ CNT (NPG) and (**b**) *L*_22_ CNT. (**c**) Thickness-dependent desalination performance in (7, 7) CNTs. The desalination parameters in (7, 7) CNT and the corresponding NPG are marked with green dashed squares and blue dashed circles for comparison.

**Figure 3 membranes-13-00525-f003:**
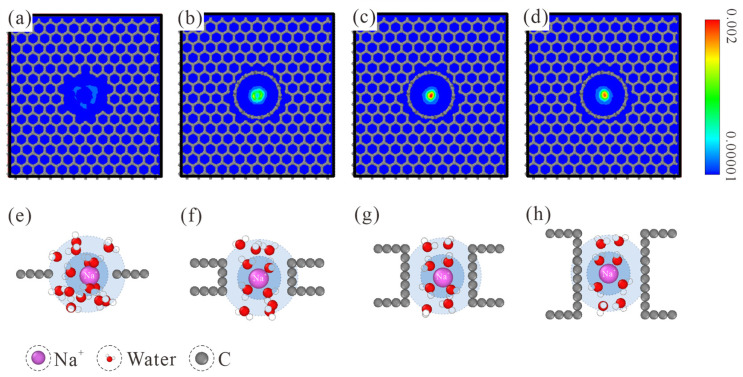
(**a**–**d**) Number density map of ion distribution within the thickness-variant (7, 7) CNTs. The corresponding CNT lengths are *L*_1_, *L*_4_, *L*_7_, and *L*_10_, respectively. (**e**–**h**) MD snapshots of Na^+^ hydration shells in *L*_1_, *L*_4_, *L*_7_, and *L*_10_ CNTs. The first and second hydration shells are marked with blue and light blue backgrounds, respectively. Only water molecules within both hydration shells are displayed.

**Figure 4 membranes-13-00525-f004:**
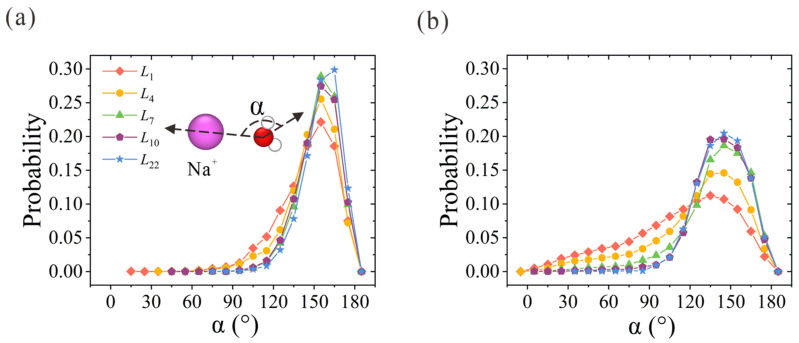
Water orientation in the (**a**) first and (**b**) second hydration shells of Na^+^ ions within thickness-variant (7, 7) CNTs. *α* is defined as the angle between the oxygen-ion vector and the water dipole vector.

**Figure 5 membranes-13-00525-f005:**
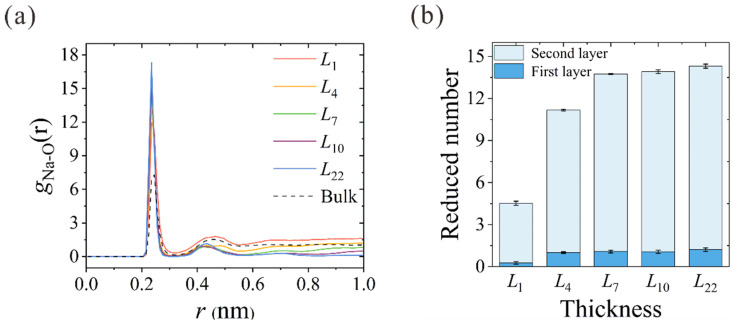
(**a**) Thickness-dependent RDFs between Na^+^ ions and oxygen atoms of water molecules (O). RDF in bulk solution is also plotted for comparison. (**b**) Reduced number of water molecules in the first and second hydration shells of different CNTs.

**Figure 6 membranes-13-00525-f006:**
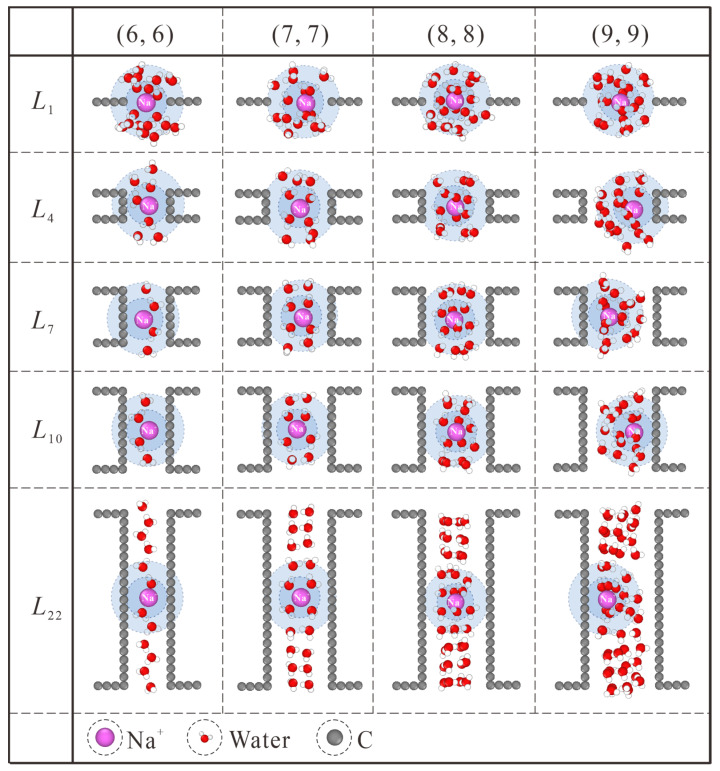
Thickness-dependent hydration configuration of Na^+^ within (6, 6), (7, 7), (8, 8) and (9, 9) CNTs. The first and second hydration shells are marked with blue and light blue backgrounds, respectively.

**Table 1 membranes-13-00525-t001:** Coordination numbers in the first (*Nc*_1_) and second hydration shells (*Nc*_2_).

	Bulk	*L* _1_	*L* _4_	*L* _7_	*L* _10_	*L* _22_
*Nc* _1_	5.6	5.2	4.5	4.4	4.4	4.3
*Nc* _2_	17.1	12.9	7.0	4.4	4.3	4.0

## Data Availability

Not applicable.

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
