# Peer review of "Competition between Hydration Shell and Ordered Water Chain Induces Thickness-Dependent Desalination Performance in Carbon Nanotube Membrane"

_membranes, 2023, doi:10.3390/membranes13050525_

Round 1
Reviewer 1 Report
The manuscript entitled "Competition between hydration shell and ordered water chain induces thickness-dependent desalination performance in carbon nanotube membrane" proposes interesting insights into
the molecular mechanisms of the thickness-dependent desalination performance in unidimensional CNT nanochannel, showing a clear finding of impact in the potential optimization of new desalination membranes.
The work is clear, well-written, consistent and provide evidence by means of MD that CNT membranes show efficient desalination performance when the length of CNT reaches a critical value that enables salt ions to form ordered hydration shells. I find that these results provide key factors of interest for experimentalists.
Therefore, I suggest its publication in the present form.
Author Response
Reviewer #1
The manuscript entitled "Competition between hydration shell and ordered water chain induces thickness-dependent desalination performance in carbon nanotube membrane"proposes interesting insights into the molecular mechanisms of the thickness-dependent desalination performance in unidimensional CNT nanochannel showing a clear finding of impact in the potential optimization of new desalination membranes.
The work is clear, well-written consistent and provideevidence by means of MD that CNT membranes show efficient desalination performance when the length of CNT reaches a critical value that enables salt ions to form ordered hydration shells. I find that these results provide key factors of interest for experimentalists.
Therefore I suggest its publication in the present form.
Our Response:
We thank the reviewer for the positive evaluation on our work.Several minor revisions have been made for the manuscript. We believe the present version is more fit for publication.
Reviewer 2 Report
This paper describes simulation efforts to optimize the water flux and salt rejection for membranes of different CNT thickness. The methods are sound and data is presented in a clear manner. I would recommend this for publication.
Lines 126-129: Please explain how you came to these values. Provide citation.
Figure 4a and 4b: Keep same scale on y-axis.
There are some minor grammatical errors in the manuscript.
Line 121-123: This sentence is framed incorrectly. Please reframe.
Author Response
Reviewer #2
This paper describes simulation efforts to optimize the water flux and salt rejection for membranes of different CNT thickness. The methods are sound and data is presented in a clear manner. I would recommend this for publication.
Comment#1: Lines 126-129: Please explain how you came to these values. Provide citation.
Our Response:
First of all, we thank the reviewer for the positive evaluation on our work. We apologize for the previous ambiguous and misleading expressions in this part. Selection of feed pressure is very important in MD simulations. For the L=25 Å (7, 7) CNT, we have tested a series of feed pressure values including 100, 200, 400 and 800 MPa. It turned out that no ion permeation was observed for the former three cases. Three Na+ and three Cl- ions were transported through the CNT membrane in the 800 MPa case. But the water flux rate in the 100 and 200 MPa cases are extremely slow. As a result, we have selected a 400 MPa feed pressure when complete salt rejection and relatively higher water flux rate can be acquired simultaneously. As for the Ref. 30 (J. Phys. Chem. C 2020, 124, 20498-20505. Pressure-dependent ion rejection in nanopores.), based on investigations on pressure-dependent ion rejection behaviors in (10, 10) CNT, it was found that the ion flux undergoes first an exponential and then a linear increase. While for the CNTs we adopted with smaller diameters such as (7, 7), when the feed pressure is 400 MPa, complete salt rejection can still be satisfied. When reducing the membrane thickness, salt rejection exhibits thickness-dependence for all the four dimeter cases. Thus, we think it is appropriate to use the 400 MPa feed pressure. We have now revised the corresponding expressions in the revised manuscript.
Comment#2: Figure 4a and 4b: Keep same scale on y-axis
Our Response:
Thanks for your kind suggestion, we have now modified the figure 4a and 4b as you suggested.
Fig. 4 Water orientation in the (a) first and (b) second hydration shells of Na+ ion within thickness-variant (7, 7) CNTs.
Comment#3: There are some minor grammatical errors in the manuscript.
Our Response:
We apologize for the grammatic errors we made in the original manuscript, we have now carefully read the manuscript and corrected them in the revised manuscript.
Comment#4: Line 121-123: This sentence is framed incorrectly. Please reframe.
Our Response:
We thank the reviewer’s careful reading of our manuscript, we have now corrected this sentence in the revision. The reframed sentence is “The first step is to set the pressures on both pistons in the feed and permeate sides to the ambient pressure of 0.1 MPa. The desalination system is then allowed to relax for 5 ns under these conditions.”
Reviewer 3 Report
The authors perform Coulombic LJ MD simulations of water transport through CNTs of varying thicknesses and diameters via a pressure difference between two graphene pistons. Na+ and Cl- ions are also present to examine salt rejection in the context of desalination membranes.
The authors demonstrate that the relationship between CNT thickness and the balance of water flux rate and salt rejection stems from the hydration configuration of the ions within the CNT. They explore the competition between the hydration shells and the ordered water chain structure within the CNTs membrane.
The analysis of the water flux rate, salt rejection, ion distribution and water orientation is scientifically sound and presented well. A mostly clear picture is painted of how water structure changes with CNT thickness both in terms of ordered chains and hydration of ions as well as how this may influence performance.
However, the concept of the "critical value" of the CNT thickness/length as discussed in the Introduction, is underrepresented in the discussion of the results and requires more attention/clarity.
When discussing results in Figure 6 on page 7, the critical thickness is said to exhibit diameter dependence and seem somewhat smaller than the reported value of 2.34 nm. Clarity surrounding this would be beneficial, particularly as the conclusion states that the salt rejection increases with thickness but levels off after a critical value. What is this critical value then?
In addition, the conclusion states the most efficient desalination performance occurred when the length reached a critical value. What is this critical value according to the results?
The increased permeate pressure compared to experimental desalination is said to increase computational efficiency. Ref. 30 warns of using high pressures when considering ion rejection simulations. Can the authors comment on the computational demands of these simulations? Arguably inexpensive methods such as LJ potentials may allow for more realistic pressure differences to be explored. In addition, the higher pressure is attributed as the reason for discrepancy from previously reported critical CNT thickness.
The quality of the English language is adequate though requires attention/minor revision before publication.
In some cases, sentences are quite lengthy and contain a lot of commas. Also, sentences need tidying e.g. page 2 "hydration state of salt ions within nanochannel" may benefit from using "the nanochannel"
Author Response
Reviewer #3
The authors perform Coulombic LJ MD simulations of water transport through CNTs of varying thicknesses and diameters via a pressure difference between two graphene pistons. Na+ and Cl- ions are also present to examine salt rejection in the context of desalination membranes.
The authors demonstrate that the relationship between CNT thickness and the balance of water flux rate and salt rejection stems from the hydration configuration of the ions within the CNT. They explore the competition between the hydration shells and the ordered water chain structure within the CNTs membrane.
The analysis of the water flux rate, salt rejection, ion distribution and water orientation is scientifically sound and presented well. A mostly clear picture is painted of how water structure changes with CNT thickness both in terms of ordered chains and hydration of ions as well as how this may influence performance.
Comment#1: However, the concept of the "critical value" of the CNT thickness/length as discussed in the Introduction, is underrepresented in the discussion of the results and requires more attention/clarity.
When discussing results in Figure 6 on page 7, the critical thickness is said to exhibit diameter dependence and seem somewhat smaller than the reported value of 2.34 nm. Clarity surrounding this would be beneficial, particularly as the conclusion states that the salt rejection increases with thickness but levels off after a critical value. What is this critical value then?
In addition, the conclusion states the most efficient desalination performance occurred when the length reached a critical value. What is this critical value according to the results?
Our Response:
We thank the reviewer for the positive comments on the scientific merit of our work, we also appreciate the reviewer for providing valuable information for improving our manuscript. We agree with the reviewer that an explicit critical value is important for the practical desalination applications. Based on our simulation results, we can summarize the critical thickness values as L1(0.34nm), L7 (1.078nm), L10 (1.447nm) and L10 (1.447nm) for (6, 6), (7, 7), (8, 8) and (9, 9) CNTs, respectively. The diameter-dependence of critical thickness value stems from the weaker confinement in large CNTs. We admit the critical thickness values provided above cannot be directly applied to the practical desalination applications. As the saturated salt rejections in (8, 8) and (9, 9) CNTs are below 60% (as shown in Fig. S2), which is inefficient and unsatisfactory. In practical desalination applications, the operation pressure is usually not higher than 10 MPa, which may loosen the restriction on the CNT diameter. It has been found that the ideal size of the CNTs for desalination applications can be as large as 1.1 nm diameter. Hence, the (8, 8) and (9, 9) CNT membranes can still be used for ideal salt rejection and a larger critical thickness value maybe expected in practical devices. This does not conflict with the reported critical thickness value of 2.34 nm. Moreover, we believe that a universal mechanism is as important as an explicit critical thickness value, as the latter is highly sensitive to the pressure condition. The critical value of 2.34 nm in Ref. 22 is the result when the pressure is 100 MPa. We believe that this critical value would be somewhat different when CNT membranes are used in practical desalination applications. By figuring out the competition between hydration shell and ordered water chain structure, we believe our work would also shed light on the design of practical desalination membranes. We apologize for our previous improper and misleading discussions and thank the reviewer again for the suggestions on improving our work. We have now revised the corresponding expressions in the revised manuscript.
Comment#2: The increased permeate pressure compared to experimental desalination is said to increase computational efficiency. Ref. 30 warns of using high pressures when considering ion rejection simulations. Can the authors comment on the computational demands of these simulations? Arguably inexpensive methods such as LJ potentials may allow for more realistic pressure differences to be explored. In addition, the higher pressure is attributed as the reason for discrepancy from previously reported critical CNT thickness.
Our Response:
We thank the reviewer for pointing out this issue, we have now revised the corresponding expressions in the revised manuscript. Selection of permeate pressure is very important in MD simulations. In spite of the use of LJ potential in MD simulations, a slightly larger pressure is usually needed to increase the computational efficiency. According to Ref. 30, they have investigated pressure-dependent ion flux in (10, 10) CNTs. A larger diameter than the one that achieves complete rejection is selected to guarantee its sensitivity of ion rejection to pressure. CNTs with smaller diameters are adopted in our work. For the L=25 Å (7, 7) CNT, we have tested a series of feed pressure values including 100, 200, 400 and 800 MPa. It turned out that no ion permeation was observed for the former three cases. Three Na+ and three Cl- ions were transported through the CNT membrane in the 800 MPa case. But the water flux rate in the 100 and 200 MPa cases are extremely slow. As a result, we have selected a 400 MPa feed pressure when complete salt rejection and relatively higher water flux rate can be acquired simultaneously. Moreover, not only for the (7, 7) CNT, but also for other three diameter cases, salt rejection exhibits sensitivity to CNT membrane thickness. Thus, we think the selection of 400 MPa feed pressure is appropriate.
Comment#3: In some cases, sentences are quite lengthy and contain a lot of commas. Also, sentences need tidying e.g.page 2 "hydration state of salt ions within nanochannel" may benefit from using "the nanochannel"
Our Response:
We thank the reviewer’s careful reading of our manuscript, we have now corrected the improper expressions in the revised manuscript